# Pterostilbene and Probiotic Complex in Chemoprevention of Putative Precursor Lesions for Colorectal Cancer in an Experimental Model of Intestinal Carcinogenesis with 1,2-Dimethylhydrazine

**DOI:** 10.3390/cancers15082401

**Published:** 2023-04-21

**Authors:** Márcio Alencar Barreira, Márcio Wilker Soares Campelo, Conceição da Silva Martins Rebouças, Antoniella Souza Gomes Duarte, Maria Lucianny Lima Barbosa, Said Gonçalves da Cruz Fonseca, Raphaela Ribeiro Queiroz, Érica Uchoa Holanda, Ana Beatriz Aragão de Vasconcelos, Vitória Jannyne Guimarães de Sousa Araújo, Gabriel Maia Diniz, Reinaldo Barreto Oriá, Paulo Roberto Leitão de Vasconcelos

**Affiliations:** 1Walter Cantídio University Hospital, Federal University of Ceará, Fortaleza 60430-140, CE, Brazil; 2Department of Surgery, School of Medicine, Federal University of Ceará, Fortaleza 60430-140, CE, Brazil; 3School of Medicine, Christus University Center (UNICHRISTUS), Fortaleza 60192-345, CE, Brazil; 4Laboratory of Tissue Healing, Ontogeny, and Nutrition, Department of Morphology, School of Medicine, Federal University of Ceará, Fortaleza 60430-170, CE, Brazil; 5Department of Pharmacy, School of Pharmacy, Federal University of Ceará, Fortaleza 60430-370, CE, Brazil

**Keywords:** chemoprevention, colorectal neoplasms, aberrant crypt foci, probiotics, pterostilbene, intestinal microbiota, intestinal inflammation, Wnt signaling pathway, carcinogenesis

## Abstract

**Simple Summary:**

The composition of the intestinal microbiota, chronic inflammation, and oxidative stress are factors related to the onset of colorectal cancer (CRC). Diet is the environmental factor most related to the development of non-hereditary CRC. Pterostilbene has great potential as an antitumor drug for CRC chemoprevention. Several strains of probiotics in multiple combinations, concentrations, and dosages have been studied for cancer prevention. The available evidence is insufficient to justify the chronic and simultaneous administration of pterostilbene and probiotics with *Lactobacillus* and *Bifidobacterium* to a population at high risk of CRC. In vivo studies may contribute to the incorporation of dietary supplementation with the ability to reduce the incidence of CRC. The combination of substances should be stimulated with a view to potentiating the expected effect through action on different targets.

**Abstract:**

Dietary supplementation with pterostilbene (PS) and/or a probiotic (PRO) may ameliorate the intestinal microbiota in disease conditions. This study aims to evaluate PS and PRO for the chemoprevention of putative precursor lesions for colorectal cancer (CRC) in an experimental model of intestinal carcinogenesis with 1,2-dimethylhydrazine (1,2-DMH). Sixty male Wistar rats were equally divided into five groups: Sham, 1,2-DMH, 1,2-DMH + PS, 1,2-DMH + PRO, and 1,2-DMH + PS + PRO. PRO (5 × 10^7^/mL) was offered in water, and PS (300 ppm) was provided in the diet ad libitum. 1,2-DMH (20 mg/kg/week) was administered for 15 consecutive weeks. In the 25th week, proctocolectomy was conducted. PRO alone and PRO combined with PS were the best intervention strategies to improve experimental 1,2-DMH-induced CRC regarding several parameters of carcinogenesis. Our findings may contribute to the development of novel preventive strategies for CRC and may help to identify novel modulators of colon carcinogenesis.

## 1. Introduction

Colorectal cancer (CRC) is the second neoplasm with the highest number of potentially preventable deaths, behind lung neoplasms. About 45.1% of cancer-related deaths in the United States are related to modifiable factors [1]. Aberrant crypt foci (ACF) are pre-neoplastic lesions associated with CRC development [2]. Most ACF are found in the distal colon segment [3], and the number of ACF is a predictive factor of the incidence of colorectal adenomas and CRC [2].

Diet is the environmental factor most closely related to the development of non-hereditary CRC [4]. It is estimated that thousands of cancer deaths can be prevented through dietary modifications [5], and up to 75% of CRC cases are associated with an unhealthy diet [6]. Many epidemiological studies have shown the relationship between adherence to the Mediterranean diet and a low incidence of CRC [7]. Natural foods with potential antitumorigenic properties are good targets for CRC prevention [8].

Pterostilbene (PS) and resveratrol are part of the stilbenes. PS, a polyphenolic compound, has a superior action to resveratrol in the attenuation of intestinal oxidative stress (OS), reducing reactive oxygen species (ROS) and regulating the mitochondrial redox state in vivo and in vitro models [9] with potential antitumoral effects to improve CRC [10]. PS has optimal oral absorption, metabolic stability, and lipophilicity [11,12]. PS can be rapidly absorbed and widely distributed in tissues with 2 to 25 times higher levels than in blood due to its high lipophilicity [13]. PS in large doses is considered pharmacologically safe [10] and has become popular due to its antioxidant and anti-inflammatory functions [12]. PS added to the diet, even at high doses (30, 300, and 3000 mg/kg/day), was found safe after a 28-day supplementation in rats [14].

The intestinal microbiota (IM) is increasingly recognized as modulating cancer-related immune and inflammatory responses [15]. Diet and environment play an important role in modulating the IM in CRC [16]. Higher numbers of *Escherichia coli*, *Escherichia faecalis, Fusobacterium nucleatum,* and *Streptococcus gallolyticus* are increased with cancer development [17]. The administration of several strains of probiotics (PRO) in multiple combinations, concentrations, and dosages has been beneficial in modulating the IM and preventing cancer [15].

In recent years, the number of studies addressing the antioxidant properties of PROs has significantly increased. The most used strains are *Lactobacillus* (L) and *Bifidobacterium* (B), which release antioxidant enzymes (SOD and CAT) and metabolites (butyrate, GSH, and folate), with immune system stimulation. In addition, those strains are shown to improve postprandial lipids involved in oxidative cell injury, with a metal chelating action, increased Nrf2 expression, and OS-reducing exopolysaccharides release [18]. The probiotic *B. animalis subsp lactis BB-12* is the most studied *Bifidobacterium strain.* It has excellent tolerance to gastric acidity and bile, optimal intestinal mucosa adherence, and pathogen inhibitory actions, and favors immunological responses [19]. *L. acidophilus* shows greater stability than *B. bifidum* in the intestinal microflora of mice in the long-term (5 months) [20]. PRO with *lactobacillus* and *bifidobacterium* has a long history of safety [21]. However, not all PROs protect against CRC development. The beneficial impact of PRO supplementation depends on the strain, dosage, intervention time, host physiology, and association with other dietary supplements [22]. A better understanding of IM dysbiosis effects on CRC pathophysiology is of utmost need [23], and its diet modulation may be a promising approach to preventing CRC [24].

Rodent carcinogenesis models are fast, reproducible, and exhibit an adenoma-to-carcinoma progression similar to that found in humans [25]. 1,2-dimethylhydrazine (1,2-DMH) is the eldest and most used carcinogen to induce tumors in rats [26]. Although the 1,2-DMH-induced carcinogenesis model does not represent the complexity of the human disease, it can be a valuable tool to study CRC and test novel chemopreventive anti-cancer agents [27]. Comparative histopathology revealed many similarities between intestinal cancer in rodents and humans, and comparative molecular pathology also showed genetic similarities [28].

The ideal and precise composition of the IM to promote antitumor effects and immune responses is not yet known [29]. The gut ecosystem in patients at higher risk of CRC may benefit from probiotic supplementation considering optimal concentrations, the duration of therapy, and the method of supplementation [30]. Although probiotics are attractive agents for the prevention of CRC, their effectiveness has not yet been fully established, and more studies need to be carried out [31].

Our study aims to evaluate the isolated or combined effect of PS and PRO supplementation on the CRC tumorigenesis markers, OS reduction, and histopathological scores after 1,2 -DMH-induced CRC. The potential protective effect of PRO and PS in preventing precursor lesions for CRC would be particularly beneficial for high-risk patients.

## 2. Materials and Methods

### 2.1. Animals

The sample consisted of 60 male Wistar rats (*Rattus norvegicus albinus*) from the Christus University Center (UNICHRISTHUS) vivarium, weighing 80 ± 10 g. The study protocol was approved by the Ethics and Animal Research Committee (nº 031/ 2018). The rats were kept in polypropylene cages in a temperature-controlled environment (22 ± 1 °C) with a 12-h light/dark cycle, free access to drinking water, and a standard chow diet. Animals that showed any signs of illness or died were excluded from the study and replaced.

### 2.2. PRO and PS Preparation

A PRO complex capsule (GNC, EUA) contains 50 billion (50 × 10^9^) colony-forming units (CFU) with the following bacteria: *Lactobacillus acidophilus* (CUL 60), *Lactobacillus acidophilus* (CUL 21), *Bifidobacterium bifidum* (CUL 20), and *Bifidobacterium animalis subsp. Lactis* (CUL 34). The PRO capsule was diluted in drinking water to reach a solution with a concentration of 5 × 10^7^ CFU/mL. An adult rat ingests approximately 20 mL (1 × 10^9^ CFU) of water daily. The solution was offered ad libitum and changed three times a week.

Eight capsules with 150 mg of PS (Mental Refreshment, EUA) were mixed with 4 kg of the standard chow diet (Nuvilab CR-1, Quintia, BRA) to produce a diet with 300 ppm of PS or 3 mg of PS/10 g of diet. The PS molecular weight was 256.3 g/mol and the PS diet concentration was 1.17 mmol/kg. A rat weighing 200 g consumes on average 10 to 20 g of the diet with 3 to 6 mg of PS/day or 15 to 30 mg of PS/Kg/day. Tests were performed to verify the presence of PS in the rats’ diet and liver (Appendix A).

### 2.3. Experimental Design

The rats were randomly divided into five groups of 12 animals (control group or Sham, cancer group or 1,2-DMH, 1,2-DMH + PS, 1,2-DMH + PRO, and 1,2-DMH + PS + PRO). The rats in the experimental group received a weekly subcutaneous injection of 1,2-DMH (D161802; Sigma-Aldrich, St. Louis, MO, USA) at a dose of 20 mg/kg body weight for 15 weeks [32]. The carcinogen was dissolved in 0.9% NaCl (pH 6.5) and the control group received the equivalent dose of 0.9% NaCl without adding the carcinogen.

After dividing the groups, solutions with PRO (in the water) and/or PS (in the diet) were administered ad libitum from the first day of the research until euthanasia during the 25th week (Figure 1).

### 2.4. Surgical Procedure and Sample Preparation

At the end of 25 weeks, the animals were anesthetized with 10% ketamine hydrochloride (80 mg/kg/weight) and 2% xylazine hydrochloride (10 mg/kg/weight). The animals were positioned in dorsal decubitus on a wooden board and immobilized by fixing their limbs. Then, a laparotomy and proctocolectomy were performed. Subsequently, the surviving animals were sacrificed by hypovolemic shock after a section of the abdominal aorta was cut. Confirmation of death occurred by verifying the absence of respiratory movements (apnea), heartbeat (asystole), and pulse.

The proctocolectomy product was opened longitudinally at the antimesenteric border for intestinal lavage and extended with the exposed mucosa. After macroscopic evaluation and division of the colon into three equal segments, the specimens were randomly stored in a 10% buffered formalin solution (*n* = 30) and in a freezer at −80 °C (*n* = 30). The samples separated for histology and immunohistochemistry were embedded in paraffin using a conventional method and stained with hematoxylin and eosin (H&E). Then, the paraffin blocks were used to make new slides that were stained with methylene blue (MB) at a concentration of 0.1% [33]. In the frozen samples, the distal segments of the colon were used to measure oxidative stress.

### 2.5. Analyzed Variables

#### 2.5.1. Microscopy

The colonic mucosa was evaluated with an optical microscope with 20× and 40× objective magnification. To evaluate the H&E-stained slides, a blind evaluator used the inflammatory score of MacPherson and Pfeiffer [34]: zero (0) for normal histological findings; one (1) for villus shortening, loss of crypt architecture, sparse infiltration of inflammatory cells, vacuolization, and edema (<25%); two (2) for villus shortening, crypt necrosis, extensive inflammatory cell infiltration, vacuolization, and edema (25% to 50%); and three (3) for villus shortening, crypt necrosis, intense inflammatory cell infiltration, vacuolization, and edema (>50%).

To evaluate the slides stained with MB, 10 fields per bowel segment (distal, middle, and proximal) were photographed randomly at 400× magnification. The crypts were analyzed in cross-sections, and the factors analyzed were the number of aberrant crypt foci (ACF) and the location in the colon (distal, middle, and proximal). ACF was considered when the crypts had at least two criteria: an increased crypt size, a thicker epithelial layer, more intense staining (due to nuclear increase and mucin depletion) [33], an increased pericrypt zone, an elliptical shape [35], and a reduction of goblet cells greater than 50% [36]. The ACF were not classified as hyperplastic and dysplastic.

#### 2.5.2. Immunohistochemistry by the Tissue Microarray Technique

Six cylindrical distal colon fragments were collected from each rat from paraffin blocks. The material was collected with a 2 mm-diameter needle (Quick-Ray UNITMA^®^, Seongnam-si, Republic of Korea) to offer good sampling, ease the construction of the recipient block, and avoid damage to the donor block. Then, the material was included in three paraffin blocks with 70 wells. In the same blocks, tissues were included to serve as a positive control for immunohistochemical reactions. Sequential 3 μm-thick sections of the tissue microarray block were deposited on silanized glass slides for conventional H&E staining and immunohistochemistry reactions.

Immunohistochemistry for iNOS, NF-kB, IL1-β, TNF-α, Ki67, P53, β-catenin, and Wnt3a proteins was performed using the streptavidin–biotin–peroxidase method [37]. In this technique, the slides were deparaffinized, hydrated in xylene and alcohol, and immersed in a retrieval solution of acid or basic pH. Then, antigenic retrieval took place for 30 min at 95 °C in an automated medium (PT-LINK). After cooling, washings were performed with Dako wash buffer solution, interspersed with blocking endogenous peroxidase with 3% H_2_O_2_ solution (20 min). The sections were incubated for 1 h with primary goat anti-KI67 antibody ab15580 (Abcam, Boston, MA, USA, 1:200), Anti-beta-catenin ab32572 (Abcam, 1:200), Wnt3a PAS37320 (Invitrogen, 1:500), P53 IS616 (Flex, Dako), Anti-NFKB p65 ab16502 (Abcam, 1:200), Anti-iNOS ab283655 (Abcam, 1:200), Anti-IL1-β ab283818 (Abcam, 1:100), and TNF-α ab307164 (Abcam, 1:100) diluted in antibody diluent. After washing in wash buffer solution, incubation was performed with HRP polymer (DAKO) for 30 min. The sections were washed again with a wash buffer, followed by staining with chromogen 3,3’diaminobenzidine-peroxide (DAB) [38] and counterstaining with Mayer’s hematoxylin. Finally, the samples were dehydrated, and slides mounted. Negative controls were processed simultaneously and incubated with serum diluent only.

The images were captured using a light microscope coupled to a camera with a LAZ 3.5 acquisition system (Leica DM1000, Wetzlar, Germany). Ten fields were photographed per histological section (40× objective), trying to select the areas with the highest marking in each animal (hot areas). For counting positive cells marked by each field, the adobe photoshop 8.0 program was used to obtain the total tissue area and the immunostained area. Positive cells were considered with brown staining within the cytoplasm for iNOS, NF-kB, IL1-β, TNF-α, and Wnt3a and within the nucleus for Ki67 and both for P53 and β-catenin. The most evident labeling of the protein was considered, avoiding the background. Then, to measure the percentage (%) of the marked area, the following calculation was performed: marked area (%) = immunomarked area (pixels) × 100/total area (pixels) [39].

#### 2.5.3. Oxidative Stress Markers

Samples from the distal segment of the large intestine were thawed and homogenized in cold EDTA (0.02 M) or KCL (0.15 M) to prepare a 10% homogenized suspension and estimate glutathione (GSH) and malondialdehyde (MDA) levels. Tissue GSH levels were estimated by the Sedlak method [40], with minor modifications. Approximately 100 μL aliquots of tissue homogenate were mixed with 80 μL of distilled water and 20 μL of trichloroacetic acid (50%, *w*/*v*) and centrifuged at 4500 rpm for 15 min. The supernatant (100 µL) was mixed with 200 µL of TRIS buffer (0.4 M, Ph 8.9) and 10 µL of 5,5’-dithiobis (2-nitrobenzoic acid) (DTNB, Sigma-Aldrich, USA). The absorbance of GSH was read at 412 nm using a control reagent (without the homogenate). The concentration was expressed in mg/g of tissue.

To determine the level of MDA in the tissues, the 2-thiobarbituric acid assay was used, which assays the level of lipid peroxidation (LP) in biological samples [41]. Aliquots of 125 μL of tissue homogenized were mixed with 750 μL of 1% H_3_PO_4_ and 250 μL of 0.6% 2-thiobarbituric acid and incubated for 1 h in a bath at 100 °C. Then, the solution was cooled on ice for 20 min, and 1 mL of n-butanol was added. The mixture was centrifuged (2000 rpm, 15 min at 4 °C), and 100 μL of supernatant was added to 96-well plates to read the absorbance at 535 nm, using a control reagent (without the homogenate). The MDA concentration was expressed in nmol/mg tissue.

### 2.6. Statistical Analysis

Statistical analysis was performed using GraphPad Prism software, version 6.0. To assess normality, the Shapiro–Wilk test was used. Parametric data were evaluated by the one-way ANOVA test of variance followed by Tukey’s multiple comparisons test, and non-parametric data were analyzed by the Kruskal–Wallis test followed by Dunns’ multiple comparisons test. The significance level adopted was 0.05 (α = 5%), and descriptive levels (p) lower than this value were considered significant. All quantitative results were expressed as mean ± standard error of the mean (SEM), except for histopathological score data and the number of ACF that were presented as median, minimum, and maximum values.

## 3. Results

Figure 2 shows the representative images of colon sections stained with H&E and MB. In the Sham group, normal crypts were observed, showing preserved goblet cells and enterocytes and the absence of inflammatory infiltrate and edema in the mucosa, submucosa, and muscle layer. In the 1,2-DMH group, a marked loss of tissue architecture and a reduction in goblet cells were seen. In addition, increased infiltration of inflammatory cells, pericrypt zone enlargement, and highly stained crypts were observed. The 1,2-DMH + PS group did not show significant improvements in histopathological scores. The PRO (1,2-DMH + PRO and 1,2-DMH + PS + PRO) groups showed more preserved goblet cells and reduced inflammatory processes. However, ACF were also observed in the PRO groups (Figure 2I,J).

Table 1 shows the average of ACF identified by field in the three segments of the large intestine. In the large intestine middle and distal segments, the 1,2-DMH group showed a higher mean ACF per field compared to the Sham group. In the distal colon segment, a reduction in the number of ACF per field was found in the 1,2-DMH + PRO group, while in the medial colon segment, the reduction of ACF per field was seen in the groups 1,2-DMH + PRO and 1,2-DMH + PS + PRO. The 1,2-DMH + PS group was not able to reduce the number of ACF per field, in any colonic segment, compared to the 1,2-DMH group. The substances used were not able to reduce the number of ACF per field in the proximal segment of the colon. The highest number of ACF per field was observed in the distal large intestine segment.

The 1,2-DMH group had a higher inflammatory score than the control group. In the middle and distal segments of the colon, the reduction in the inflammatory score was evident in the 1,2-DMH + PRO and 1,2-DMH + PS + PRO groups. The reduction in the inflammatory score was more intense in the distal segment of the colon. The 1,2-DMH + PS group was not able to reduce the inflammatory score, in any colonic segment, in relation to the 1,2-DMH group. The tested compounds were not able to reduce inflammatory scores in the proximal segment of the colon (Table 2).

MDA levels were higher in the 1,2-DMH group compared to the Sham group. 1,2-DMH + PRO and 1,2-DMH + PS + PRO supplementation significantly reduced MDA levels in the distal large intestine. In addition, GSH levels were significantly reduced in the 1,2-DMH group, an effect that was improved by either compounded 1,2-DMH + PS + PRO supplementation (Figure 3).

The immunostaining of iNOS, NF-kB, TNF-α, and IL1β proteins is present in all groups of the experiment (Figure 4A). A statistically significant difference was found between the 1,2-DMH group and the Sham group regarding the expression of iNOS, NF-kB, TNF-α, and IL1β proteins. TNF-α and IL-1β immunolabeling in the treatment groups was not different from the Sham. The immunoexpression of iNOS and NF-kB proteins was reduced in the 1,2-DMH + PRO and 1,2-DMH + PS + PRO groups compared to the 1,2-DMH group. The 1,2-DMH + PS group was not able to significantly alter the expression of iNOS, and NF-kB proteins in relation to the 1,2-DMH group (Figure 4B,C). The immunoexpression of TNF-α and IL1β proteins was reduced in the 1,2-DMH + PS, 1,2-DMH + PRO. and 1,2-DMH + PS + PRO groups compared to the 1,2-DMH group (Figure 4D,E).

The immunostaining of Ki67, P53, β-catenin, and Wnt-3a proteins is more evident in the intestinal crypts and present in all groups of the experiment (Figure 5A). There is a statistically significant difference between the 1,2-DMH group and the Sham group regarding the expression of Ki67, P53, β-catenin, and Wnt-3a proteins. The immunoexpression of Ki67 and P53 proteins was reduced in the 1,2-DMH + PS, 1,2-DMH + PRO, and 1,2-DMH + PS + PRO groups in relation to the 1,2-DMH group (Figure 5B,C). The immunoexpression of β-catenin and Wnt-3a proteins was reduced in groups 1,2-DMH + PRO and 1,2-DMH + PS + PRO in relation to group 1, 2-DMH. The 1,2-DMH + PS group was not able to significantly alter the expression of β-catenin and Wnt-3a proteins in relation to the 1,2-DMH group (Figure 5D,E).

## 4. Discussion

The concept of cancer prevention is to delay, regress, or eliminate precancerous lesions. The ACF number can be used as a reliable biomarker of preneoplastic and cancerous lesions in the large intestine [42]. This information contributes to developing different preventive strategies for CRC and helps in identifying modulators of colon carcinogenesis [35]. As expected, our study confirmed a higher number of ACF in the 1,2-DMH-induced CRC group compared to the Sham rats. The PRO and PS-PRO combination improved the number of microscopic lesions, including the number of ACF.

The histopathological inflammation scoring by MacPherson and Pfeiffer has been used in experimental research to quantify the intensity of the acute inflammatory process of the colon [34]. However, we showed an interesting association between the number of ACF and the intensity of the inflammatory process after 1,2-DMH-induced CRC. Our data indicated that both PRO and the PRO and PS combination improved the inflammatory score, indicating that this score is a valuable tool to assay CRC histopathological remediation.

Furthermore, our data support prolonged inflammation being associated with higher levels of ROS and CRC severity [43]. GSH is one key antioxidant of the tissue arsenals for balancing OS. Importantly, reduced levels of GSH in the 1,2-DMH-induced CRC were found in our study and were closely associated with increased intestinal MDA (a surrogate marker of OS). Zińczuk et al. observed a reduction in GSH levels and an increase in MDA levels in the blood of patients with CRC compared to healthy controls [44]. ACF from CRC patients were compared to healthy tissues, confirming a pro-oxidative crypt milieu. Only PRO and PS combined supplementation could improve intestinal GSH and MDA levels in our cancer-challenged rats. Of note, *B. lactis* A6 was found to attenuate OS by lowering MDA levels and increasing GSH levels in colonic tissues. It may also attenuate the inflammatory response via the downregulation of TNF-α, IL-1β, and IL-6 in colonic tissues [45]. Intestinal MDA levels have been consistently related to tumor invasion depth and lymph node metastasis [44].

Our data showed that PS and PRO alone or in combination improved TNF-α and IL-1β immunolabeling, two critical pro-inflammatory cytokines related to CRC pathogenesis [46], in the colon tissue compared to the 1,2-DMH-challenged controls, reaching the level of the sham group. Sustained high levels of TNF-α and IL-1β may precede putative CRC precursor lesions [47].

Although our findings with PS improved Ki67, p53, IL-1β, and TNF-α intestinal tissue expression, it did not improve Wn3a and β-catenin expression. Interestingly, a low dose of PS (40 ppm) in the diet of rats for 45 weeks lessens colon tumorigenesis with reduced Wnt/β-catenin signaling and with cyclin D1 expression inhibition (a well-known Wnt downstream effector) [48]. This discrepancy may be due to different time periods of PS supplementation between both studies. Corroborating our data, PS was also found to reduce the expression of proinflammatory cytokines TNF-α (by 51%) and IL-1β (by 47.7%) in mucosal scrapings derived from the colon of the Azoxymethane (AOM)-injected rats [48].

Interestingly, we found some P53-positive cells showing cytoplasmatic labeling, especially within the colonic crypt cells. Jansson and colleagues have highlighted that P53 cytoplasmatic accumulation could indeed be related to tumorigenesis and that both nuclear and cytoplasmatic expression of P53 could be associated with poor cancer prognosis [49]. Increased P53 colonic expression is an early genetic event in the process of CRC tumorigenesis. P53 mutations have been detected in the colonic mucosal tissue even with a negative or low-grade dysplasia and later surrounding regions of high-grade dysplasia and carcinoma. The mutation of a single P53 allele leads to tumoral growth advantage, with tumor cell clonal expansion with disrupted cellular DNA repair mechanisms [50]. It is recognized that such mutations may occur in the absence of any morphological changes [51]. Neoplasia may arise within different populations of cells in separate areas of the same colon [52].

Cytoplasm and nuclear β-catenin expression is a useful marker for premalignant lesions of rat colon carcinogenesis [53,54]. In our study, we found β-catenin immunolabeling in colonic crypts, even in the ones with no clear ACF alteration. Early cytoplasmic or nuclear localization of ß-catenin may be a meaningful proxy of CRC precursor lesions that may corroborate with the more laborious genetic analyses [55].

Notably, a diet with added PS (40 ppm) for eight weeks was found to suppress ACF (57% inhibition) and reduce iNOS expression [56]. In another study, when PS was added to the diet (50 or 250 ppm) of mice for 6 to 23 weeks, PS reduced the number of ACF which was more pronounced when the supplementation was given for a longer time, with diminished iNOS and COX-2 levels and declines of Wnt/β-catenin signaling, VEGF, and cyclin D1 expression, along with high apoptosis rates in the colon [57]. Previously reported studies had intestinal carcinogenesis induced with the carcinogen AOM. There are no reports of experimental studies with the addition of PS in the diet to prevent the development of CRC.

PS was recognized to modulate IM with anti-inflammatory effects in an experimental study with dextran sodium sulfate (DSS)-induced colitis. PS attenuated the severity of colitis by reducing IL-2 and IL-6 and shifted the IM composition toward a healthier profile, increasing the concentration of *Bifidobacterium* and reducing the concentration of harmful bacteria in the gut [58].

Our beneficial findings with PS could be mainly linked to pinostilbene, the primary metabolite of PS in the colon. Pinostilbene may play important roles in the anti-colon cancer effects elicited by orally administered PS [59]. Pinostilbene bio-availability may be influenced by the IM through demethylation [60]. PS-deconjugated metabolites are biologically more active than conjugated ones [61]. The combination of PRO with PS may favor a composition of microbiota that facilitates the metabolization of pinostilbene, thus improving its anti-cancer effects that may not be seen with PS alone.

In support of our data, the chemopreventive effects of PRO Dahi (20 g/day) with two types of *Lactobacillus* (2 × 10^9^ CFU/g *Lactobacillus acidophilus* LaVK2 and *Lactobacillus plantarum* Lp9) have been highlighted following intestinal carcinogenesis (IC) with four applications of 1,2-DMH (40 mg/kg body weight) in Wistar rats [62]. In addition, pre-conditioned rats (for 1 week) with a diet containing 0.2% or 4% lyophilized cultures of *L. acidophilus* were later challenged with AOM (5 mg/kg/week) for two weeks to induce IC and kept on the same diet for 10 weeks and showed a significantly lower number of ACF compared to non-supplemented controls [63].

Another study has shown that PRO with *L. acidophilus* (2 × 10^9^ CFU), orally administered (three times a week), significantly reduced the incidence and multiplicity of ACF after the induction of CRC with 1,2-DMH [64]. The preventive administration of PRO (07 CCM7766) with *L. plantarum* to rats that received applications of 1,2-DMH was able to reduce the inflammatory process in the colon through the negative regulation of proinflammatory cytokines (IL-2, IL-6, IL-17, and TNF-α,), the elevation of the anti-inflammatory cytokine IL-10, and the suppression of NF-κB activity in mucosal cells [65]. Furthermore, a PRO composed of *L. casei* BL23 improved CRC induced in mice by a single application of AOM (8 mg/kg) followed by four cycles of DSS (2.5%) in drinking water. PRO had an immunomodulatory effect through a reduction of the IL-22 cytokine and an antiproliferative effect measured through the upregulation of caspases 7 and 9. There were no significant differences between IFN-γ, TNF-α, IL-6, and IL-10. None of the mice in the group that received *L. casei* BL23 developed macroscopic tumors, while 67% of the mice in the cancer group did. Ki-67 immunolabeling was lower in the group that received PRO [66].

It has been documented that *B. lactis* inhibits the NF-kB pathway and regulatory genes in intestinal epithelial cells after the induction of IC with AOM and DSS [67]. The impairment of the intestinal epithelial barrier can be considered one of the first events that occurs in intestinal inflammation since it facilitates the entry of antigens from the intestinal lumen to the mucosa, which can lead to an uncontrolled and exacerbated immune response [68].

An experimental model of CRC induction with 1,2-DMH (30 mg/kg, twice a week for three weeks) investigated the effect of a diet supplemented with a combination of *L. casei*, *B. bifidum* (10^8^ CFU/mL) and sphingomyelin (0.05%) and showed improvements in the number of ACF in the colon [69]. Another study showed that long-term (24 weeks) consumption of PRO (*B. longum* and *L. gasseri*) resulted in significant inhibition of ACF formation after the induction of CRC with 1,2-DMH [70].

In addition, PRO with De Simone formulation (*L.casei*, *L. plantarum*, *L. bulgaricus*, *L. acidophilus*, *B.longum*, *B. breve*, *B. infantil*, and *Streptococcus thermophilus*) showed anticancer and anti-inflammatory activity in an experimental induction model of the CRC [71]. PRO enriched in *L. delbrueckii* UFV-H2b20 or *B. animalis* var. lactis Bb12 resulted in a reduction in the total number of ACF (55.7% and 45.1%, respectively). However, a synergistic effect of *Lactobacillus* and *Bifidobacterium* supplementation on ACF inhibition was not observed [72]. *L. acidophilus* has been found to have a more important effect than *B. bifidum* in the chemoprevention of CRC. The administration of PRO inhibited the incidence of colonic lesions by about 57% with *L. acidophilus* and 27% with *B. bifidum* compared to the group that had the induction of IC with AOM and did not receive PRO [73]. Future studies are warranted to better balance the PRO composition of selected bacteria with more anti-tumoral effects.

Table 3 summarizes the main selected studies with PRO, discussed in this article. All of the studies are from experimental CRC models. The current literature supports the benefit of PRO to intestinal health in improving colonic histopathological scores, OS, and inflammation biomarkers.

The limitations of this study are that we could not dissect in more depth the OS metabolism (including SOD and catalase tissue levels) and the downstream Wnt signaling and other regulatory cell proliferation pathways. In the future, it would be interesting to evaluate IM modulation by PS and/or PRO.

Figure 6 summarizes our findings following the 1,2-DMH-challenge in the rats’ colons and the protective effects of PS and/or PRO supplementation.

Progression from this precursor lesion to CRC is a multistep process, accompanied by alterations in several suppressor genes that result in abnormalities of cell regulation, and it has a natural history of 10–15 years. The 10–15-year time frame of this process provides an opportunity for both primary and secondary prevention [74]. An advantage of chemoprevention over the current secondary prevention strategy of routine colonoscopy is the potential to intervene early in the carcinogenic sequence to reduce CRC risk, allowing for less frequent surveillance exams and a reduction in the number of invasive cancers.

## 5. Conclusions

New combination therapies are desirable for CRC prevention and adjuvant treatment. This was the first study to evaluate both PS and PRO supplementation in the chemoprevention of precursor lesions for CRC. Further studies should be performed to clarify which PRO-selected microorganism composition and concentration are more synergistic to improve PS biological effects and CRC chemoprevention.

Altogether our findings suggest that PRO alone and PRO combined with PS were the best intervention strategies to improve experimental 1,2-DMH-induced CRC regarding several parameters of carcinogenesis. Our findings highlight the importance of preventive measures to control intestinal microbiota and minimize CRC incidence in high-risk populations.

We acknowledge, due to the relatively low number of results obtained and experiments performed, we could not dissect in-depth the fine downstream inflammatory and early tumorigenesis crosstalk pathways (including the canonical Wnt signaling) that could shed light to find novel and promising pharmacological targets to halt CRC precursors lesion progression. Such results may guide future clinical trials in large populations worldwide to prevent/slow the occurrence of CRC.

## Figures and Tables

**Figure 1 cancers-15-02401-f001:**
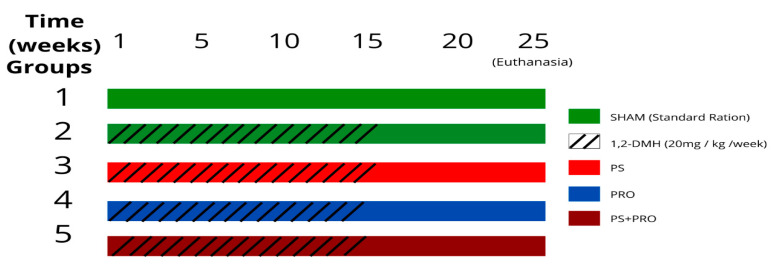
Experiment design.

**Figure 2 cancers-15-02401-f002:**
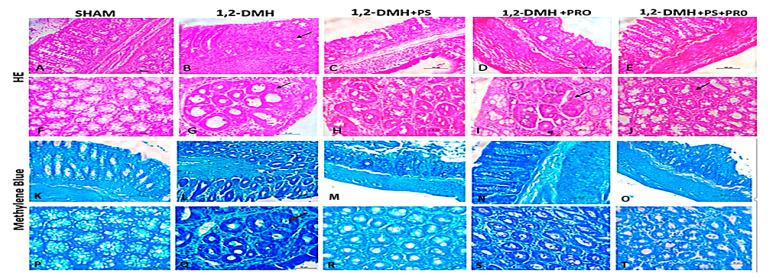
Histopathology of the experimental groups. Photos 2 (**A**–**E**) and 2 (**K**–**O**) were under 20× objective magnification (scale bar 100 μm) and the longitudinal section. Photos 2 (**F**–**J**) and 2 (**P**–**T**) were under 40× objective magnification (scale bar 50 μm) and the cross-section. Black arrow—the presence of an aggregate of ACF with more intense staining due to nuclear enlargement and mucin depletion.

**Figure 3 cancers-15-02401-f003:**
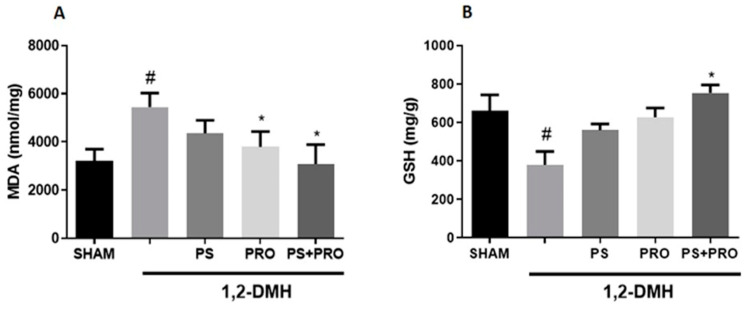
Analysis of oxidative damage by measuring MDA and GSH levels in the distal colon segment. (**A**) MDA and GSH (**B**) levels in the distal colon segment. The values are presented as the mean ± SEM. # *p* < 0.05 vs. Sham and * *p* < 0.05 vs. 1,2-DMH group. For statistical analysis, the one-way ANOVA test was used followed by Tukey’s post-test.

**Figure 4 cancers-15-02401-f004:**
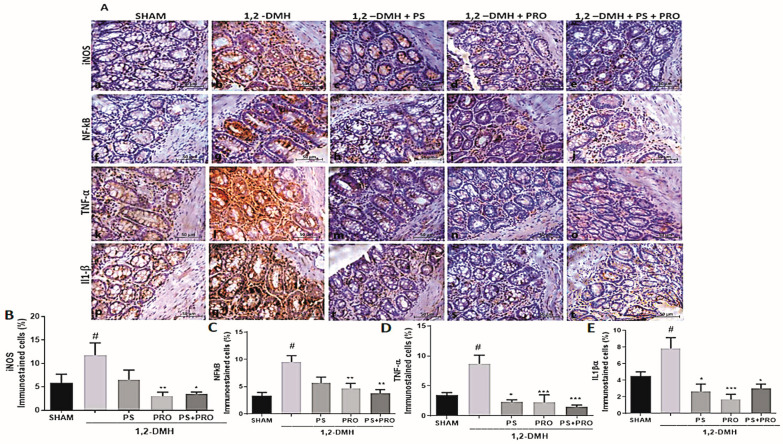
Immunoexpression of iNOS, NF-kB, TNF-α, and IL1β proteins in the distal colon segment. (**A**) Representative histology of iNOS, NF-kB, TNF-α, and IL1β immunolabeling in the experimental groups. The graphs (**B**–**E**) represent the mean ± SEM of the percentage of the immunopositive area for iNOS, NF-kB, TNF-α, and IL1β in relation to the total area. # *p* < 0.05 vs. Sham and * *p* < 0.05, ** *p* < 0.01, *** *p* < 0.001 vs. 1,2-DMH group. (**B**,**D**) The data were analyzed using the Kruskal–Wallis test followed by the Dunn’s post-test. (**C**,**E**) The data were analyzed by the one-way ANOVA test followed by Tukey’s post-test.

**Figure 5 cancers-15-02401-f005:**
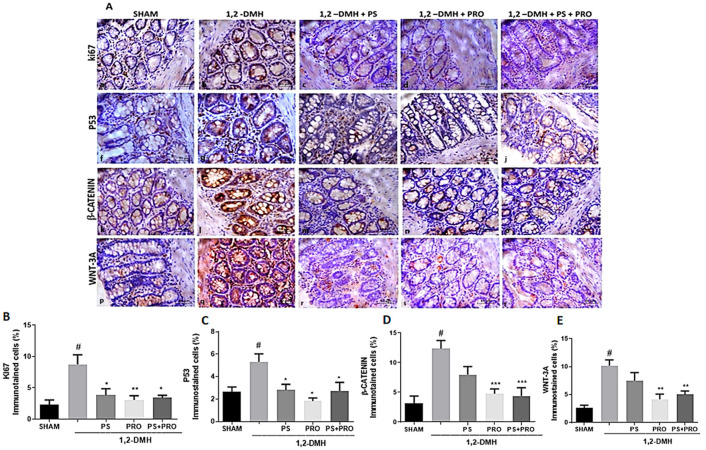
Immunoexpression of Ki67, P53, β-catenin, and Wnt-3a proteins in the distal colon. (**A**) Representative histology of Ki67, P53, β-catenin, and Wnt-3a protein immunolabeling in the experimental groups. The graphs (**B**–**E**) represent the mean ± SEM of the percentage of the immunopositive area for Ki67, P53, β-catenin, and Wnt-3a in relation to the total area. # *p* < 0.05 vs. Sham and * *p* < 0.05, ** *p* < 0.01, *** *p* < 0.001 vs. 1,2-DMH group. (**B**,**C**) The data were analyzed using the Kruskal–Wallis test followed by the Dunn’s post-test. (**D**,**E**) The data were analyzed by the one-way ANOVA test followed by the Tukey’s post-test.

**Figure 6 cancers-15-02401-f006:**
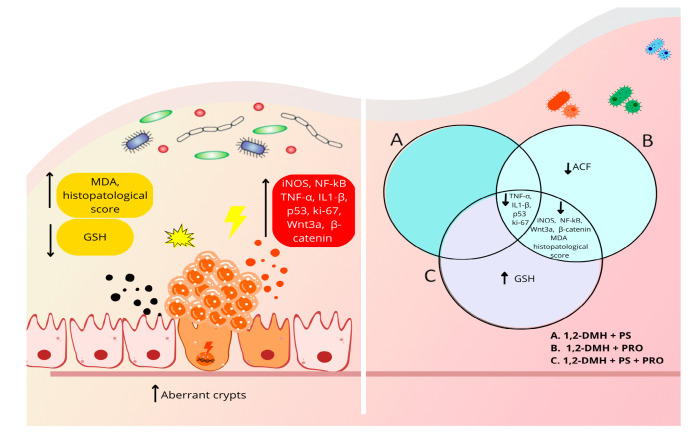
The outcome of PRO and/or PS supplementation in an experimental model of CRC development with 1,2-DMH. 1,2-DMH increased histopathological scores, OS (elevated MDA and reduced GSH), and selected inflammatory and tumorigenesis markers (Ki67, β-catenin, Wnt3a, p53, NF-kB, iNOS, IL1-β, and TNF- α). 1,2-DMH + PRO and 1,2-DMH + PS + PRO reduced histopathological scores, and selected inflammatory and tumorigenesis markers (Ki67, β-catenin, Wnt3a, p53, NF-kB, iNOS, IL1-β, and TNF- α) and improved OS (MDA). 1,2-DMH + PS + PRO also increased intestinal GSH levels. 1,2-DMH + PS reduced p53, Ki67, IL1-β and TNF-α tissue expression, but without improving OS and histopathological scores. ↑, increased; ↓, decreased.

**Table 1 cancers-15-02401-t001:** Number of ACF per field in the three colonic segments. The values are expressed as the median, minimum, and maximum, where # *p* < 0.05 vs. Sham and * *p* < 0.05 vs. group 1,2-DMH. The data were analyzed using the Kruskal–Wallis test followed by the Dunn’s post-test.

Intestinal Segments	Experimental Groups				
	Sham	1,2-DMH	1,2-DMH + PS	1,2-DMH + PRO	1,2-DMH + PS + PRO
Proximal	0.0 (0–0.1)	1.8 (0.5–3.0)	0.5 (0–2)	0.5 (0–2)	0.5 (0–1)
Medium	0.0 (0–1)	1.9 (1.4–3.0) #	1.0 (0–2.8)	0.126 (0–1.4) *	0.25 (0–1) *
Distal	1.1 (0.3–2)	4.3 (3–6) #	2.25 (1–5)	2.0 (0.5–3.33) *	3.0 (1–5)

**Table 2 cancers-15-02401-t002:** Histopathological score of inflammation in the three colonic segments. The scores are expressed as the median, minimum, and maximum values, where # *p* < 0.05 vs. Sham and * *p* < 0.05, ** *p* < 0.01 vs. 1,2-DMH group. The data were analyzed using the Kruskal–Wallis test followed by the Dunn’s post-test.

Intestinal Segments	Experimental Groups				
	Sham	1,2-DMH	1,2-DMH + PS	1,2-DMH + PRO	1,2-DMH + PS + PRO
Proximal	1 (0–1)	3 (1–3) #	2 (1–3)	2 (1–3)	2 (1–2)
Medium	0 (0–1)	2 (1–3) #	2 (0–2)	1 (1–2) *	1 (0–2) *
Distal	0 (0–1)	3 (2–3) #	2 (1–3)	1 (1–2) **	2 (1–2) **

**Table 3 cancers-15-02401-t003:** Summary of animal model studies of PRO supplementation following CRC induction.

Studies	Bacteria Strains	Concentration	Supplementation Time	Benefits
Wang et al., 2022 [45]	*B. lactis A6*	4 × 10^9^ CFU/day	3 weeks	↓ MDA, ↑ SOD, GSH, and ↓ TNFα, IL-1β and IL-6 levels in colon tissues
Kim et al., 2010 [67]	*B. lactis*	5 × 10^9^ CFU/g	9 weeks	↓ NF-kB and COX-2 expression
Mohania et al., 2013 [62]	*L. acidophilus* and *L. plantarum*	2 × 10^9^ CFU/g	32 weeks	↓ Number of ACF
Rao et al., 1999 [63]	*L. acidophilus*	diet containing 0.2% or 4% lyophilized cultures	10 weeks	↓ Number of ACF
Chang et al., 2012 [64]	*L. acidophilus*	2 × 10^9^ CFU/mL	10 weeks	↓ Number of ACF
Štofilová et al., 2015 [65]	*L. plantarum*	1 × 10^9^ CFU/mL	28 weeks	↓ Pro-inflammatory cytokines (IL-2, IL-6, IL-17, and TNF-α,), NF- κB, COX-2, and iNOS proteins; ↑ goblet cell
Jacouton et al., 2017 [66]	*L. casei BL23*	5 × 10^9^ CFU/mL	46 days	↓ Ki-67 immunolabeling
Marzo et al., 2022 [69]	*L. casei* and *B. bifidum*	1 × 10^9^ CFU/mL	66 days	↓ Number of ACF
Foo et al., 2011 [70]	*L. gasseri* and *B. longum*	1 × 10^11^ CFU/g and 5 × 10^9^ CFU/g	24 weeks	↓ Number of ACF
Bassaganya-Riera et al., 2012 [71]	*L.casei, L. plantarum, L. bulgaricus, L. acidophilus, B.longum, B. breve, B. in-fantil,* and *Streptococcus thermophilus*	1.2 billion bacteria per mouse/day	68 days	↓ Adenoma and adenocarcinoma formation; ↑ mRNA expression of TNF-a
Liboredo et al., 2013 [72]	*L. delbrueckii* or *B. lactis*	3 × 10^8^ CFU/mL	14 weeks	↓ Number of ACF (55.7% vs. 45.1%, respectively).
Agah et al., 2019 [73]	*L. acidophilus* or *B. bifidum*	1 × 10^9^ CFU/g	5 months	↓ Incidence of colonic lesions (57% vs. 27%, respectively), CEA, and CA19-9

↑, increased; ↓, decreased

## Data Availability

The data presented in this study are available in this article.

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
