# Peer review of "Pterostilbene and Probiotic Complex in Chemoprevention of Putative Precursor Lesions for Colorectal Cancer in an Experimental Model of Intestinal Carcinogenesis with 1,2-Dimethylhydrazine"

_cancers, 2023, doi:10.3390/cancers15082401_

Round 1

Reviewer 1 Report (New Reviewer)

Done

Author Response

Dear reviewer,

We appreciate Reviewer 1 acceptance of the paper, as the reviewer did not add any new comments or suggestions. We did an extensive English proof review.

Regards,

                                                                         Márcio Alencar Barreira

Reviewer 2 Report (New Reviewer)

I congratulate the authors for their idea, study design and results, in this first study to evaluate supplementation with both pterostilbene and a complex of probiotics, in chemoprevention of putative precursor lesions for colorectal cancer, in rats. I see that the authors already corrected their manuscript, following the Editor’s and the Reviewer’s comments. I have listed some suggestions for consideration, below:

1. Abstract: I would suggest to summarize in the Conclusion the main findings of your study (in one sentence), before writing: "Our findings.."

2. Keywords: It would be advisable to use other Keywords, not those belonging to the title. This would increase the likelihood of the paper being found by readers. The importance of Keywords is to improve indexing. Only “Aberrant Crypt Foci” do not belong to the title.

3. Figure 2: Please correct to “1,2-DMH + PS, 1,2-DMH + PRO, and 1,2-DMH + PS + PRO”. “D” is missing.

4. Please insert place of Fig. 4A in the main text.

5. Discussion: This paragraph is beautifully conceived. Please:

a. Replace “VSL # 3” with “De Simone Formulation”, given the potential conflicts and to be scientifically and deontologically correct (https://www.probiotixx.info/en/a-major-decision-by-us-court-ruled-against-vsl3/, https://onlinelibrary.wiley.com/doi/full/10.1111/apt.14515, https://www.visbiome.com/pages/vsl-3-story).

b. Summarize all mentioned studies with probiotics in a table, plus add some more recent ones.

c. “One of limitation” – correct to “limitations” and insert all of them.

6. Conclusion is short and crispy, as it should be. Only please insert text about Figure 6 and Figure 6 per se, in Discussion, above. Please also delete “e” (from “TNF-α, Il-1 β, p53 e ki-67”) (Figure 6).

Thank you.

Author Response

Dear reviewer,

The manuscript has been revised again according to the new comments. We appreciate the time and effort that you dedicated to providing feedback on our manuscript and we are grateful for the insightful comments and valuable improvements to our paper. We have considered the comments and tried our best to address each one of them. Those changes are highlighted within the manuscript.

Reviewer 2

  1. Abstract: I would suggest to summarize in the Conclusion the main findings of your study (in one sentence), before writing: "Our findings.."

Authors reply: We modified the abstract, as requested. See page 1, lines 34 and 35.

  1. Keywords: It would be advisable to use other Keywords, not those belonging to the title. This would increase the likelihood of the paper being found by readers. The importance of Keywords is to improve indexing. Only “Aberrant Crypt Foci” do not belong to the title.

Authors reply: We added four new keywords (Intestinal microbiota, Wnt signaling pathway, Intestinal inflammation, Carcinogenesis). See page 2, line 40.

  1. Figure 2: Please correct to “1,2-DMH + PS, 1,2-DMH + PRO, and 1,2-DMH + PS + PRO”. “D” is missing.

Authors reply: We corrected as requested. See updated figure 2.

  1. Please insert place of Fig. 4A in the main text.

Authors reply: We added that as requested. See page 8, line 295.

  1. Discussion: This paragraph is beautifully conceived. Please:
  2. Replace “VSL # 3” with “De Simone Formulation”, given the potential conflicts and to be scientifically and deontologically correct (https://www.probiotixx.info/en/a-major-decision-by-us-court-ruled-against-vsl3/, https://onlinelibrary.wiley.com/doi/full/10.1111/apt.14515, https://www.visbiome.com/pages/vsl-3-story).

Authors reply: We appreciate the reviewer’s compliment about our paper discussion. We corrected the minor points raised by the reviewer as requested. See page 12, line 443.

  1. Summarize all mentioned studies with probiotics in a table, plus add some more recent ones.

Authors reply: We have added a new paragraph and table. See page 12 and 13, line 454-462.

  1. “One of limitation” – correct to “limitations” and insert all of them.

Authors reply: We corrected it. See page 13, line 464.

  1. Conclusion is short and crispy, as it should be. Only please insert text about Figure 6 and Figure 6 per se, in Discussion, above. Please also delete “e” (from “TNF-α, Il-1 β, p53 e ki-67”) (Figure 6).

Authors reply: As requested, Figure 6 has been moved to the discussion and corrected accordingly. See page 12 and 14, lines 469-474.

All changed text was highlighted in yellow in the paper to facilitate the reviewer work. We appreciate yours comments which helped to improve further our manuscript. We look forward to hearing your thoughts.

Regards,

                                                                         Márcio Alencar Barreira

Reviewer 3 Report (New Reviewer)

The manuscript by Barreira et al. aims to demonstrate the chemopreventive effects of a polyphenolic compound, pterostilbene (PS), and those of a probiotic complex (PRO) in an in vivo model of intestinal carcinogenesis. However, the Authors fail in their intent. This because:

- The paper is well written but scientifically poor for being work carried out on animal models and by such many Authors. Molecular biology techniques such as PCR or Western Blotting are absent, no mechanism of action was hypothesized for the tested compounds, nor were the involved pathways described. Despite to several targets of inflammation (i.e., NF-κB, TNF-a) or apoptosis (i.e., p53) were detected, they did not correlate with each other.

- No analysis of the composition of the intestinal microbiota (IM) was performed to explain its potential modulation underlying the chemopreventive effects observed for the probiotic complex.

- The results in the treated groups did not always show such a significant impact compared to the group exposed to 1,2-DMH (i.e., Figure 3).

- Differences in cytosolic and nuclear expression of a target (such as β-catenin) are not captured in the work or were not always evaluated (such as for p53). In addition, the immunohistochemistry results deserve to be confirmed with further different analysis technique.

- The combination of PS and PRO was not sufficiently investigated, for example by showing a possible synergistic or additive effect, nor it was reported a significant efficacy advantage of combination over the use of the individual compounds.

- Considering the low number of results obtained and experiments performed, it is rather bold by the Authors to state in the Conclusions that “Our findings highlight the importance of preventive measures to control intestinal microbiota and minimize the CRC incidence in high-risk populations”.

Therefore, the article deserves substantial improvement and cannot be considered for its publication.

Author Response

Dear reviewer,

The manuscript has been revised again according to the new comments. We appreciate the time and effort that you dedicated to providing feedback on our manuscript and we are grateful for the insightful comments and valuable improvements to our paper. We have considered the comments and tried our best to address each one of them. Those changes are highlighted within the manuscript.

Reviewer 3

1.The paper is well written but scientifically poor for being work carried out on animal models and by such many Authors. Molecular biology techniques such as PCR or Western Blotting are absent, no mechanism of action was hypothesized for the tested compounds, nor were the involved pathways described. Despite to several targets of inflammation (i.e., NF-κB, TNF-a) or apoptosis (i.e., p53) were detected, they did not correlate with each other.

Authors reply: We acknowledge that further studies are needed, and we plan a second paper with to deepen the subject with more molecular biology protocols, including the use of PCR or Western Blotting. Unfortunately, it is not possible to use these techniques in this work. Although we understand the limitations, we believe that our paper brings already an important message at this level to guide future studies.

2.No analysis of the composition of the intestinal microbiota (IM) was performed to explain its potential modulation underlying the chemopreventive effects observed for the probiotic complex.

Authors reply: We are planning to do this for another study. We will put it as a suggestion at the end of the discussion. See page 13, line 466-467.

3.The results in the treated groups did not always show such a significant impact compared to the group exposed to 1,2-DMH (i.e., Figure 3).

Authors reply: We obtained level of significance for key early carcinogenesis biomarkers and for GSH and MDA levels, as oxidative stress proxies, regarding PS and PRO groups. More studies are needed to extend and reinforce those findings. 

  1. Differences in cytosolic and nuclear expression of a target (such as β-catenin) are not captured in the work or were not always evaluated (such as for p53). In addition, the immunohistochemistry results deserve to be confirmed with further different analysis technique.

Authors reply: We justified some immunohistochemical findings related to the expression of two important proteins (P53 and ß-catenin) related to the development of CRC. See page 10, lines 371-387. More studies are needed to extend and reinforce those findings. We are planning this for a next paper.

  1. - The combination of PS and PRO was not sufficiently investigated, for example by showing a possible synergistic or additive effect, nor it was reported a significant efficacy advantage of combination over the use of the individual compounds.

Authors reply: The association needs to be better studied in the next experiments. Figure 6 shows in detail the results achieved with the treatment groups used.

  1. - Considering the low number of results obtained and experiments performed, it is rather bold by the Authors to state in the Conclusions that “Our findings highlight the importance of preventive measures to control intestinal microbiota and minimize the CRC incidence in high-risk populations”.

Authors reply: We agree with the reviewer and altered the conclusion adding the following statement: “We acknowledge, due to relative low number of results obtained and experiments performed, we could not dissect in-depth the fine downstream inflammatory and early tumorigenesis crosstalk pathways (including the canonical Wnt signaling) that could shed light to find novel and promising pharmacological targets to halt CRC precursors lesion progression. Such results may guide future clinical trials in large populations worldwide to prevent/slow the occurrence of CRC.” See page 14, lines 494-499.

All changed text was highlighted in yellow in the paper to facilitate the reviewer work. We appreciate yours comments which helped to improve further our manuscript. We look forward to hearing your thoughts.

Regards,

                                                                         Márcio Alencar Barreira

Reviewer 4 Report (New Reviewer)

The work sent to me for review has already been assessed and reviewed. This is my first time reviewing it and that's why I look at it in two separate ways. Firstly, as a new, previously unknown article, which I read as a new, previously unknown text. Secondly, the corrections made to its previous version, marked in yellow in the current copy of the article.

In the first of the mentioned aspects, I believe that the reviewed article meets all the criteria of a well-prepared scientific research work. It was well planned in its initial stage, properly fulfilled and correctly described. All experimental data are well documented and their results statistically analyzed, also in a way that does not raise any objections. The authors did a very good idea of placing part of the experimental information in the supplementary materials. This allowed to keep a much better and more compact form of the work itself. The evaluated article corresponds very well to the detailed profile of the "Cancers" journal.

In the second aspect of the assessment mentioned above, I also see only positive features of the introduced changes. They were really needed, and by adapting the work to the earlier suggestions, it clearly gained in the quality and clarity of the message. According to the authors “cover letter”, practically all the suggested changes were introduced in the corrected article text, and additionally, a few of them were addressed in a very matter-of-fact way by expressing their comments. The introduced changes make the evaluated work fully correct and corresponding, in its substantive form, to all the requirements of your journal.

There are still minor spelling and editorial errors which should be removed. I suppose, probably, it will be corrected during the preparation of the text for printing.

For my part, I recommend accepting the work for publication in its current form, without any further corrections.

Author Response

Dear reviewer 4,

We appreciate Reviewer 4 acceptance of the paper, as the reviewer did not add any new comments or suggestions. We appreciate the comments. Thank you for taking the time to evaluate our work.

Regards,

                                                                         Márcio Alencar Barreira

Round 2

Reviewer 3 Report (New Reviewer)

The Authors did not comply with the reviewer's requests and made no significant changes to the manuscript, which remains scientifically poor to be considered for publication

This manuscript is a resubmission of an earlier submission. The following is a list of the peer review reports and author responses from that submission.

Round 1

Reviewer 1 Report

The manuscript entitled “Pterostilbene and probiotic complex in chemoprevention of 2 colorectal cancer in an experimental model of intestinal 3 carcinogenesis with 1,2-dimethyl-hydrazine” by Márcio Alencar Barreira et al.,  was talks about the chemopreventive efficacy of Pterostilbene and probiotic complex against DMH induced colon cancer. The experiment was well planned and executed, that was evidenced by photographical representations. However, authors should need to concentrate more in their discussion part. If possible, additional parameters will support the results

Abstract

Page 1, line numbers 27-30: if possible rewrite the sentences to make it more understandable

IM

Figure 2, Q is that 40X magnification? recheck

Results

If possible add tumor size data in table

Is there any data available with you regarding the bacterial populations in the colon?

DMH, being carcinogen produces free radicals,

SOD and CAT are front line defense mechanism against free radical’s attack, Authors need to justify, why missed those to parameters in their assay

Discussion

Reduce the discussion part

Add some details about the structure – relationship activity of Pterostilbene

The whole discussion part occupied with other research studies, rather than the current study. Authors should take care of this part and need to discuss their own research.

Reviewer 2 Report

The manuscript of Barrerira et al. evaluates the effect of a polyphenol (pterostilbene) and probiotic (containing Lactobacillus and Bifidobacterium) on the development of colorectal cancer. The experiments are well-designed and conducted, and the measurements of different markers of inflammation and oxidative stress provide information on the effects of these compounds. However, several aspects of the manuscript have to be improved before being considered for publication.

Major comments

1. Experimental deseing, lines 130-135. How was 1,2-DMH dissolved? The solvent used should be specified since it is usually dissolved in EDTA 1 mmol/L, pH 6.5, and the control animals received only the solvent.

2. The authors indicated that PS was mixed in the pellet to produce a diet with 300 ppm. Could the authors estimate the amount of PS consumed by the rats. In addition, to indicate de dose in ppm, it would be helpful for the reader to also indicate this quantity either in mmol/kg or mg/kg,

3. How did the authors select the dose of PS? Could the dose be translated to humans, and be achieved with the diet, or it has to be provided as a supplement? The same can be applied to the probiotic.

4. The authors administered to the rats not only pterostilbene but also a probiotic with bacteria, to evaluate the effect on the microbiota of the animals, which could help a better interpretation of the results, and also to assess the role of the intestinal microbiota on CRC.

5. In the microscopy assay, lines 170-176. The authors described the determination of ACF, why they did not follow the criteria of Bird (1987)?

6. Discussion, lines 346-347. The authors stated that the anticancer effect of PS is superior to that of resveratrol in CRC in cell culture. The authors did not provide any data on concentrations of either PS or resveratrol to support such a strong statement. I recommend rewriting the sentence to provide information that could support what it is said. In addition, it cannot be compared, the results found in vivo with those of cell culture.

7. Discussion, lines 366-368. “Chiou et al. [58] mostraram re-366 dução de ACF após a inclusão de PS (50 ou 250 p.p.m) na dieta de camundongos por 6 e 367 23 semanas, com menor número de ACF nos grupos com maior concentração de PS (250 368 p.p.m).” There is text written in Portuguese, I recommend revising the English of the entire document.

8. Lines 452-456. The authors speculate on the metabolites of PS in the colon. Could the authors measure the amount of PS reaching the colon and its metabolites?

9. Lines 463-465. The authors indicate that diets rich in polyphenols and PRO constitute a mechanism of great potential in the chemoprevention of CRC. It is a already known statement, that to be included at the end of the discussion it should be supported with data on the doses that could exert this chemopreventive effect.

10. Could the authors link the results obtained for the different markers with the significant pathways included the MAPK, p53, Wnt, Jak-STAT, which have been implicated in the development of CRC. That could improve the discussion of the manuscript

Minor comments

1. Replace “1,2-dimethyl-hydrazine” by “1,2-dimethylhydrazine” all throughout the manuscript

2. Line 19. Replace “Pteriostilbene” by “Pterostilbene”

3. Line 62. Replace “reverastrol” by “resveratrol”

4. Line131. Replace “câncer” by “cancer”

5. In general, the manuscript has to be revised to check typographic as well 

Round 2

Reviewer 1 Report

The manuscript entitled “Pterostilbene and probiotic complex in chemoprevention of colorectal cancer in an experimental model of intestinal carcinogenesis with 1,2-dimethylhydrazine” by Márcio Alencar Barreira was well revised and presented. The topic was well chosen and needed for the present-day society. The authors provided appropriate responses to the queries raised by the reviewer to make it more concise. Histological grading with percentage was much appreciated.  In addition, the quality of the revised manuscript was improved by authors input.

Minor query

1.       Figure 4,  TNF-α and IL-1β, shows more inflammation in Sham group (Carcinogen Untreated) than treated groups- Need to be justified.